# Digital Technology and Services for Sustainable Agriculture in Tanzania: A Literature Review

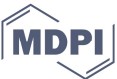

**Gilbert E. Mushi [1,2,*], Giovanna Di Marzo Serugendo [2] and Pierre-Yves Burgi [3]**

[1] Department of Informatics and Information Technology, Sokoine University of Agriculture, Morogoro P.O. Box 30007, Tanzania

[2] Computer Science Center, University of Geneva, 1205 Geneva, Switzerland; giovanna.dimarzo@unige.ch

[3] Division Système et Technologies de l'Information et de la Communication, University of Geneva, 1205 Geneva, Switzerland; pierre-yves.burgi@unige.ch

*  Correspondence: gilbert.mushi@sua.ac.tz; Tel.: +41-779823851 or +255-753147427

**Abstract:** Digital technology has the potential to eradicate extreme poverty and food insecurity to the majority of smallholder farmers in the world. This paper aims to identify knowledge gaps on digital technology for sustainable agriculture and assess their availability to smallholder farmers worldwide. The particular case of Tanzania receives special attention. We conducted an extensive literature search from relevant databases for review. The advanced digital technology in agriculture, mostly used by large scale farmers, significantly contributes to sustainable agriculture. However, the existing digital services for smallholder farmers lack sustainability in the agriculture context and hardly meet the needs for a comprehensive set of services in a complete farming cycle. In most developing countries, Tanzania case included, digital technology and  services respond to a challenge at a particular stage of the farming process or to a specific value chain. Based on this literature review, we identify inequalities among large and small farmers, as well as environmental challenges caused by ICT itself. To conclude we provide suggestions for improvements for smallholder farmers: developing a digital platform that addresses smallholder farmers' challenges in a complete farming cycle, bringing together the stakeholders at a country level, in order to achieve sustainable agriculture and support adoption of cutting-edge digital technology.  These suggestions will be the starting point for future research.

**Keywords:** digital technology; sustainable agriculture; smallholder farmers; ICTs services; precision agriculture; smart farming; farmers services; Tanzania

## 1. Introduction

The application of digital technologies in agriculture may eradicate extreme poverty and hunger in a yet constantly growing population—from 2019 to 2050 the population will increase by 2 billion people [1]. In recent years, digitization has changed the way the society performs its social-economic activities particularly due to increased interconnections through the internet and affordable digital devices creating a global digital ecosystem [2]. Digitization is increasingly becoming an essential tool of production, business and services to recover the society from unexpected novel corona virus pandemic that has brought devastating impact on the social, economic and environmental aspects [3]. The use of digital technology has proved useful in various sectors worldwide, such as Malaysian industries [4], healthcare [5] or manufacturing [3]. In the agriculture sector, digital technology has increased profitability, enhanced the quality of the products and somehow preserved the environment [6]. The current "Industry 4.0 digital transformations" apply advanced technology in the agricultural field for a more precise and real-time decision making in farming activities [7]. This new era of digital technology in agriculture uses knowledge from different disciplines, which include information science, computer and

software engineering, environmental science, remote sensing, geographical positioning systems (GIS), crop and soil science and global positioning systems (GPS) [8]. The farm management system uses modern technologies such as artificial intelligence (AI), sensors, the Internet of Things (IoT), satellite images to collect data, big data and machine learning, contributing to higher productivity and profitability in this sector [6].

However, most small and medium-sized farmers cannot afford to adopt such modern technology for sustainable agriculture, which is contrary to the United Nations Sustainable Development Goals (SDG) principle of "leaving no one behind" [9]. Smallholder farmers generate enormous employment and income worldwide while producing over 70% of the world's food needs [10]. In Tanzania, the agriculture sector is the backbone of the economy with 26.7% of the GDP, employing more than 80% of the population, and women constitute 60% of the farm workforce [11,12].

Many scientists and organizations have used different approaches to enable digital technology by smallholder farmers to increase productivity and income. Efforts include developing mobile and online services that allow smallholder farmers to access various services such as weather information, farming information and knowledge, market information, and reliable buyers for their products [13,14]. According to Boyera and Grewal [15], and Gray et al. [16], digital technology and digital farmers profiling platforms for smallholder farmers could help farmers access essential services and improve productivity. However, despite all those efforts, the sustainability of these projects remains a significant challenge for achieving sustainable agriculture [10]. Furthermore, the application of digital technology requires the study of the value chain to meet the needs of services in the context of the farmer ecosystem.

This paper is part of a larger research project to harness digital technology for sustainable agriculture in Tanzania, which aims to identify knowledge gaps on digital technology and services available to smallholder farmers and sustainability in agriculture. Moreover, it suggests digital solutions for smallholder farmers towards sustainable agriculture in developing countries. The subject aligns with the United Nations SDGs, such as eradicating poverty and hunger, sustainable cities and communities, climate action and reducing inequality [9]. Developing new digital comprehensive artifacts could solve the existing problems of digital exclusion of smallholder farmers, such as access to credit, farming knowledge, farm inputs, government services and control, and the market for their products [17–20]. Responsible agriculture actors could adopt the artifact according to their country context. Therefore, this review addresses the following questions:

1.  What digital technology and services are available to support the agriculture sector?
2.  What is the relationship between digital technology and sustainable agriculture? How do smallholder farmers fit in?
3.  What is the state-of-the-art use of digital technology and services by smallholder farmers in Tanzania?
4.  What challenges need to be addressed in relation to the above questions?

The last question concerns the future research agenda and will be further developed in a subsequent publication. In this paper we focused on digital technologies and services in agriculture, with a specific emphasis on smallholder farmers and sustainability.

We organized this paper as follows. Section 2 describes the authors' methods to select papers for this review. Section 3 reviews related works that answer the above first three questions, which guides this review. Section 4 responds to the fourth question; it analyzes and synthesizes gaps regarding the availability of digital technologies and sustainable agriculture (as defined in this paper) to smallholder farmers. Section 5 concludes with a summary of the review and suggests future work.

## 2. Research Methods

We used PRISMA guideline in this study [21], which is a standard protocol and an evidence-based framework for doing systematic review studies. We conducted an extensive literature search based on a complex query in the Web of Science (WoS), IEEE Xplore and related databases (Food and Agriculture Organization, Google Scholar and Research4Life). The aim was to find and review the latest literature in digital technology and sustainable agriculture in relation to smallholder farmers. The researchers combined the following keywords using the Boolean operators ("AND" and "OR") and parentheses during the search: digital technology, ICT services, smart farming, precision agriculture, digital farmer profiling, smallholder farmers and sustainable agriculture. The final search string was "('digital technology' OR 'ICT services' OR 'precision agriculture' OR 'smart farming' OR 'digital farmer profiling') AND 'sustainable agriculture' AND 'smallholder farmers'". However, the search string could not yield good results to FAO database due to type and differences in functionality. We conducted a search in October and November 2021, obtained and imported a total number of 1981 articles to Mendeley Desktop reference manager software (https://www.mendeley.com).

We applied exclusion criteria to the obtained results to identify relevant papers in digital technology, smallholder farmers and sustainable agriculture. We restricted the obtained results to the year of publication from 2015 to 2021 to get the latest articles in the subject area. We filtered out duplicated papers (using duplicate function in Mendeley software), articles without full text and not written in English. The inclusion criteria were as follows: (i) modern digital technologies in agriculture (e.g., smart farming, digital farmer services) including the sustainability components (economic, environmental and sustainability of the ICTs infrastructure and resources) and (ii) availability of the technology to smallholder farmers. Finally, we selected a total of 36 articles: 24 articles on global literature (21 for recent digital technologies and sustainable agriculture, three on general digital service platforms developed for smallholder farmers) and 12 for the Tanzanian case.

We separately searched the literature in the Tanzanian case in local repositories (Sokoine University of Agriculture Institutional Repository), WoS and Google Scholar. In this search, we did not limit the literature by the year of publication to obtain more detailed background information in the country's ICTs and smallholder farmers' services. We obtained 18 articles from local repositories for analysis as the result of the complex query "digital technology" OR "ICT services" AND "smallholder farmers" OR "agriculture" AND "Tanzania". We selected 12 articles for the review after filtering five articles which were similar to articles from Google Scholar and WoS, (see Table 1).

**Table 1.** Reviewed literature under PRISMA guideline.

| Search Category | Identification | | Screening | | Included |
|---|---|---|---|---|---|
| General literature | Records identified from databases (N = 1981) | Duplicate removed (N = 85) | Records screened (N = 1687) | Records excluded (N = 1581) | Studies included in review (N = 24) |
| | | Removed for other reasons (N = 209) | Reports sought for retrieval (N = 106) | Reports not retrieved (N = 11) | |
| | | | Reports assessed for eligibility (N = 95) | Reports excluded by the study criteria (N = 71) | |
| Tanzanian case | Records identified from databases (N = 18) | Duplicate removed (N = 5) | Records screened (N = 13) | Records excluded (N = 1) | Studies included in review (N = 12) |
| | | Removed for other reasons (N = 0) | Reports sought for retrieval (N = 12) | Reports not retrieved (N = 0) | |
| | | | Reports assessed for eligibility (N = 12) | Reports excluded by the study criteria (N = 0) | |

## 3. Results

We present the results of this paper in response to the research questions. First, the results of the digital technology and services available to support the agriculture sector worldwide. Second, the results of the relationships between digital technology and sustainable agriculture, focusing on smallholder farmers inclusion in digital transformation. Furthermore, we re-defined sustainable agriculture in the context of this paper to address the identified gaps in existing literature. Finally, the results of the Tanzania case current status in the use of digital technologies in agriculture and challenges towards sustainable agriculture.

### 3.1. Digital Technology and Services in Agriculture

For a long time, the agriculture sector has embraced new technologies to increase production and profitability while improving the environment. The Organization for Economic Co-operation and Development (OECD) defines digital technologies as: "ICTs (information communication technologies), including the Internet, mobile technologies and devices, as well as data analytics used to improve the generation, collection, exchange, aggregation, combination, analysis, access, searchability and presentation of digital content, including for the development of services and apps" [22].

Farmers use digital technologies in different domains of agriculture (summarized in Table 2). These domains include digital technology for farm management, financial services, market services, and farming knowledge and information services. Additionally, some digital platforms provide all essential services to farmers in the farming ecosystem. Many ICTs projects for farmers at the country level offer solutions to a particular farming problem, mainly for a specific value chain.

#### 3.1.1. Farm Management

The current industry 4.0 digital transformation in agriculture integrates IoT, cyber-physical systems, AI, Big Data, Machine Learning and Cloud computing with agricultural machinery [23]. It is more common to precision agriculture whereby innovative ICT solutions and IoT components such as sensors monitor spatial and temporal variability in farm production [7,24]. Site-specific farm management provides an understanding of soil and crop characteristics unique to each field, thus enabling farmers to apply farm inputs (such as irrigation, fertilizers, pesticides and herbicides) in small portions where needed for the most economical production [25]. Controlled farm inputs increase farm productivity and profitability and conserve the environment, promoting sustainable agriculture development [26]. Precision agriculture and smart farming rely on data management to make valuable decisions. The embedded digital technology components can be categorized into three phases: (1) data collection (IoT), (2) data management and analysis, and (3) decision making and variable rate technology (actuation) [6].

Data collection—IoT

IoT in agriculture uses sensors—devices used to collect data from the field for easy monitoring of the crops and other digital tools to collect essential data for profitable decision-making in farming [6]. The sensors are mounted in the mobile farm machinery or fixed in the field, such as a local weather station. For instance, Kilin [27], used a network of automated stations in the vineyards to detect areas affected by pathogens for site-specific application of pesticides. The stations collect real-time data such as airborne particles, temperature and relative humidity of the air and soil, solar irradiance, spores, and leaf humidity. AI is then used to analyze the spatio-temporal heterogeneity data based on optical particle counters (OPC) to identify areas affected by the pathogen (i.e., *Plasmopara viticola*) [27]. The results allow farmers to apply pesticides in specific field zones leading to cost-effective, healthy products and environmentally friendly farming practices. Saiz-Rubio [6], classified sensors into three: remote sensing, aircraft, and proximal sensing. Remote sensing, most often satellites, has been an essential tool for collecting field data in

smart farming. The satellites used to provide agricultural data include WorldView 2 and WorldView 3 multispectral satellite sensors using Normalized Different Vegetation Index (NDVI) standard [28,29]. Furthermore, the European Sentinel 2 satellite system, which gives access to 10 m 4-band multispectral data for "NDVI imagery of soil and water, covers the Earth every 10 days; the American Landsat satellites provide spectral data from the Earth each 16 to 18 days" [6,29].

Aircraft sensing, usually "remotely-piloted aircraft (RPA) and unmanned aerial vehicles (UAV)" such as drones, capture field data at a closer distance of up to 100 m, contrary to the order of 700 km of satellites [29]. Although aircraft sensing is expensive and requires high skills to generate quality field data, they are flexible and reach field areas where other equipment cannot. Proximal sensing is the latest technology based on "autonomous ground systems", promising new agriculture transformation [2]. According to Saiz-Rubio [6], in comparison to remote and aircraft sensing, proximal sensing monitors the crop in the ground at less than 2 m between a crop scanned and sensor. The payload of sensors is placed in ground vehicles that move around the field to collect accurate and quality data from the crops. Proximal sensing allows a real-time application, such as applying fertilizer where needed and spraying herbicides and pesticides where weeds or pests have been detected [30].

Robotic technology in farming is another area of interest and part of proximal sensing where unmanned ground vehicles (UGV) collect data and manage various farm activities [31]. The farmers use UGVs for soil analysis, seeding, transplanting, harvesting and crop scouting. Thus, UGVs allow a continuous field data collection process to monitor crop status and growth conditions [32]. VineRobot and Vinescount, funded by the European Commission, are examples of robotic technologies in smart farming that monitors vineyards by collecting data from the vines' canopy and creating water and nutrition status maps [6]. Industries manufacturing agricultural tools are also producing scouting robots. For example, Rowbot Systems LLC of USA introduced a multitask robotic platform to map crop growth zones, apply fertilizer and other related tasks [33]. Another example is the robot Oz the autonomous weeding and seeding [34].

Data management and analysis

A digital system receives data from different IoT devices and helps generate meaningful information for production. Large scale and commercial farmers use farm management information systems (FMIS) to acquire data, store, analyze and manipulate data in precision and smart farming. FMIS enables farmers to manage various farming activities from the initial planning stage to harvest and record important information of the performed activities [35]. Farmers can extract information such as field maps to determine crop and field conditions necessary for actions related to minimal use of resources, compliance with standards, and quality of agriculture production. There are different FMIS on the market (most are proprietary) with various features to manage farm generated data. The systems manage farm operations based on data acquired and processed automatically for planning, monitoring, supporting decision-making and keeping valuable records [36]. Hrustek [7], mentioned that FMIS records critical information, including "harvests and yields, profits and losses, farm task scheduling, weather prediction, soil nutrients transport and field mapping". A few examples of FMIS are ADAPT, Agrivi, Agroptima, Farmleap, owned mainly by companies from developed countries. More advanced FMIS provides early warning, financial management and integrates other actors such as input suppliers and product distributors.

Decision-making and variable rate applications

Farmers need to decide on the vast volume of collected data, considering different field parameters. Managing such complex data manually is difficult, time-consuming and possible for ineffective decision-making. [7]. Hrustek [7], added that farmers could use artificial intelligence (AI) and machine learning to support decision-making in agriculture

through available big data. Wolfert [37], argued that agriculture has many areas for applying different AI technologies. For instance, Giusti and Marsili-Libelli [38] developed a decision support system (DSS) based on fuzzy logic to manage irrigation considering the soil characteristic and type of crop. Additionally, Bazzani [39] developed a decision-support system (DSS) that analyzes short- and long-term availability of water based on soil type, machinery and irrigation systems. Furthermore, Rupnik et al. [40], developed AgroDSS cloud-based DSS that allow farmers to upload data or integrate with FMIS through an application programming interface (API) to get different output decisions such as farm pest management.

The variable rate technology (VRT) has made it possible for the decision to be made autonomously. According to Hrustek [3], actuation is the execution of activities in the field following decision making from collected data. VRT includes robots used to perform different farm activities (farm preparation, planting, pest and weed control, fertilization, harvesting) previously conducted by human labor or conventional farm machines [28,38]. The variable-rate device receives commands from a computerized DSS. It performs various farming tasks such as applying fertilizer, pesticides and herbicides in the specific field zones where needed (real-time applications) and harvesting [41]. A few examples of VRT machines include the automated yield monitoring system II (AYMS II) made of unique "eye" color cameras and real-time kinematics-GPS for wild blueberry harvesting [42]. A sensor-based variable rate nitrogen fertilizer (VRNF) measures nitrogen with a multispectral sensor and fertilizer spreader mounted on a tractor, for real-time application conforming to the measured nitrogen in the crop [43]. The CLAAS VRT is used to apply nitrogen fertilizer, compatible with the "ISARIA" sensor [26]. VRT increases production and preserves ecological balance through efficient farm inputs, i.e., less crop fertilizer and chemicals [44]. Figure 1 presents the three main categories of smart farming data life cycle.

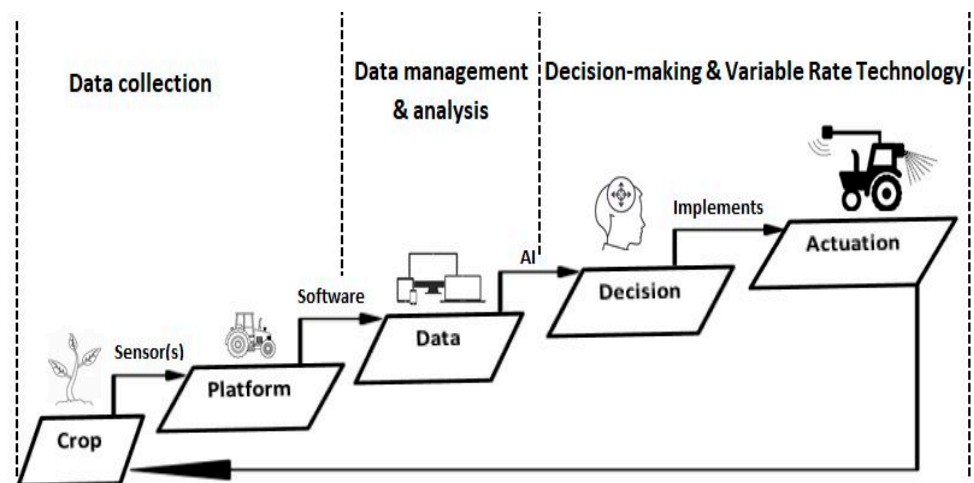

**Figure 1.** Smart farming data life cycle inspired by Saiz-Rubio and Rovira-Más [6].

### 3.1.2. Financial Services

Smallholder farmers face the challenge of access to financial services affecting agriculture production and income of many rural communities in developing countries [17]. Digital technology is an essential tool for improving access to finance and the commercialization of smallholder agriculture. A study on the awareness and use of m-banking (mobile banking) shows that most smallholder farmers in Kenya use the technology to access finance for agriculture-related activities [45]. Kirui [45], concluded that m-banking enables smallholder farmers to access investment capital for purchasing quality seeds, farm machinery, fertilizer and pesticides, leading to increased production and income. The Association for People of Haryana (AFPOH) is an ICT-based agriculture initiative in India that enables most smallholder farmers to access finance for improved agriculture

[46]. Different countries embrace digital technology to allow the commercialization of smallholder agriculture as poverty alleviation and food security strategy.

Furthermore, agriculture insurance is an essential service for smallholder farmers. The farmers normally encounter various production and market risks which lower their income and ability to produce year after year. Hess and Hazell [47], mentioned natural disasters such as extreme droughts, floods, hurricanes and pest outbreaks are common risks for smallholders. The risks cause severe impacts in economic development which leads to extreme poverty. In the past, governments and organizations designed several insurances to help small farmers towards sustainable agriculture. However, agricultural stakeholder and organizations considers index-based agricultural insurance as more effective for smallholder farmers in developing countries [48,49]. Still, majority smallholder farmers particularly in Africa have no access to insurance. For instance, approximately 650,000 farmers have access to insurance in Africa out of around 40 million smallholder farmers in Sub-Saharan African alone. [47]. The current trend of climatic change requires financial investment for agriculture transformation, including increasing availability and access to credit and insurance by smallholder farmers [50].

### 3.1.3. Knowledge and Information Services

Dissemination of agriculture information and knowledge is a critical move towards improved farming. Most smallholder farmers lack farming information and knowledge, so they rely on friends, family, and experience, resulting in low production [51]. Access to data in a complete farming cycle, from farm preparations, inputs, finance, harvesting and market of the products, creates high value in the commercialization of smallholder agriculture. Ali et al. [52], examined the critical information needs of farmers in Pakistan and developed a digital solution to deliver weather forecasts, pesticides and fertilizer information. E-agriculture initiatives in India emphasize disseminating information to most rural smallholder farmers through ICT, including management information systems, knowledge management systems and expert systems [46]. Sanga et al. [53], developed an information dissemination system to enable smallholder farmers to access critical farming information and knowledge from experts, bridging the gap of extension services through ICT. Scientists and organizations have developed mobile applications to disseminate different crops and livestock information. For instance, Kenya Agricultural and Livestock Research Organization (KALRO) has produced more than fourteen mobile applications for crops and livestock to help farmers access information and adopt modern farming techniques for increased production [54].

### 3.1.4. Market Services

Most large scale farmers use advanced FMIS, which provide linkage to critical services, including the market [55]. For instance, we can mention the combination of different methodologies to design information integration in the Netherlands for information sharing that supports the food supply chain—a movement of food into various stages from farmers to consumers and movement of money paid for the food by the consumers back to the farmers via the same steps in the reverse direction [56]. Wolfert et al. [56] argue that big data in smart farming is appealing as farmers can either be part of the closed, proprietary systems or of an open, collaborative system. A proprietary system is a highly integrated system of stakeholders bounded by terms and conditions. In contrast, with "open, collaborative systems" farmers are free to choose any stakeholder as business partners in a food supply chain. In either of the two (closed or open) scenarios, the food supply system enables farmers to exchange information with other actors in a supply chain (two-way traffic), harnessing essential knowledge for production based on consumer needs and other factors in the supply chain.

Smallholder farmers face the challenge of market access for their products [55,57]. Intermediaries force farmers to sell their products at a low price, resulting in unprofitable

production. Thanks to ICT, smallholder farmers can access market information and participate in better-paying agricultural production. Market access is one of the critical components in e-agriculture initiatives in India. Rural farmers are linked to the market and get fair prices, improving income and sustainable life [46]. ICT related cases in Africa include "eSoko" in Ghana, "Tru Trade" in Uganda and "mFarming" in Kenya, Ghana and Tanzania [58]. These programs address the challenge of access to market information and fair price for smallholder farmers' products.

Furthermore, Nigeria's "E-Wallet Scheme" enables smallholder farmers to access subsidized inputs through mobile phones. Meanwhile, "E-Krishok and Zero Hunger" in Bangladesh and "Farmes' Advisory Information System" in Tanzania provides extension services to farmers, mainly advising farmers on farm input products [53,55,58–60]. These and many other related efforts not included in this paper are promising ICT initiatives for smallholder farmers access to the market.

### 3.1.5. e-Government Services in Agriculture

Governments play a fundamental role in developing any economic sector, including agriculture. For a long time, most governments have provided various services in agriculture, most often through extension agents responsible for linking with farmers [61]. However, several limitations to using extension agents include the difficulty of reaching the many smallholder farmers scattered throughout the rural areas, the inability to deliver multiple agriculture services to farmers and the high involved costs [62]. Governments have a central role of monitoring, controlling and bringing together agricultural stakeholders for services deliverance at a single access point; thus, promoting digital technology for sustainable agriculture at a country level. OECD [22], mentioned that ICT promotes government transparency and accountability to the community. Therefore, e-government provides opportunities for the government to deliver multiple, coordinated and timely services under one roof through a network of agricultural actors. Ntaliani et al. [63] assessed the potential of e-government in the agricultural sector which suggests that government should use the e-government model to offer services to farmers and rural communities. The Indian government, through the ministry of agriculture, supports various ICT programs for smallholder farmers to access essential services such as farm inputs, financial services, subsidies and market for increased production and income [46].

### 3.1.6. Digital Farmer Profiling Platforms and Services

Apart from precision agriculture and smart farming, many ICT services provide isolated solutions packages to farmers' needs. Digital farmer profiling is a business model developed in the past few years to provide essential solutions to smallholder farmers' needs. The platform service manages farmers' data based on blockchain technology to allow farmers to share their data with other stakeholders (such as credit and insurance companies) [16]. Digital farmers profiling seems promising in service delivery to the smallholder farmers. Studies in Africa, Asia and Latin America show how digital farmers profiling enables smallholders to access essential services such as financial services and marketing of their products [10,15,64,65]. Service providers manage the data (for a fee) on behalf of other actors, including the farmers. Despite the long debate over who owns the data (between service providers and farmers), Grameen Foundation—as experienced experts in the farmer profiling platform business model, has stated that sustainability of the project is a significant challenge once the project fund ends [16]. In addition, Boyera and Grewal [15] concluded that each country and value in the crop or livestock chain would have its approach to implementing a farmer profiling platform.

**Table 2.** A summary of digital services for farmers.

| | Services | Digital Artifact Solutions | Sources |
|---|---|---|---|
| Farm management | IoT | Sensors: Fixed position, UAV, Satellites, UGV | [27,28,30,31,33] |
| | Data Management and Analysis | Farm Management Information Systems (FMIS) | [7,35,36] |
| | Decision-making and Variable Rate Technology | Variable rate nitrogen fertilizer (VRNF), CLAAS VRT, Automated yield monitoring system II (AYMS II), fuzzy logic DSS, AgroDSS | [37,38,40,43] |
| Financial services | | Index-based agricultural insurance, AFPOH, M-Banking | [45–49] |
| Knowledge and information | | Weather forecasts, pesticides, and fertilizer information; KALRO mobile applications, Farmers Advisory Systems | [52,54,60] |
| Market | | eSoko, Tru Trade, E-Wallet Scheme, E-Krishok and Zero Hunger | [46,55,58,60] |
| e-government | | Online Fertilizer Recommendation System (OFRS) in Bangladesh, AFPOH in India, KALRO in Kenya | [46,54,66] |
| Profiling platform | | Digital farmer profiling platform | [10,15,16] |

Source: Author's compilation.

### 3.2. Digital Technology and Sustainable Agriculture

According to Bhakta et al. [41], Giray and Catal [64], sustainable agriculture refers to agricultural practices that ensure long-term increased farm production and farmers' income while protecting the environment. Precision agriculture and smart farming present a high level of sustainability using the most cutting edge technology to control farm inputs such as fertilizers, irrigation, herbicides and pesticides [6]. Farmers apply farm inputs to only parts of the field that need, thus improving product quality, reducing input cost, increasing productivity, preserving the environment, and achieving economic and environmental sustainability [7,64]. Social sustainability in agriculture results from economic and ecological sustainability, whereby, refers to the availability of enough food for all people, animals, and plant species in the world [7]. Literature on sustainable agriculture mainly focuses on agricultural operations and business models for increased profit while minimizing the use of agrochemicals to promote a healthy environment and higher production quality. The new "fog computing model" is useful for a clean environment in smart agriculture. Unlike cloud computing, the fog computing model reduces carbon emissions through energy-efficient digital hardware and renewable energy resources since data are processed closer to where it is collected [65].

In addition to previous sustainability approaches, this paper focuses on the fundamental component of sustainable agriculture in the digital era: the sustainability of infrastructures and resources that support digital agriculture services for smallholder farmers. Thus, this paper categorizes sustainable agriculture into three main topics: (i) sustainability of the infrastructure and resources offering digital services, (ii) economic sustainability—long-term increased productivity and profitability, and (iii) environmental sustainability—conservation ecology and minimizing ICT pollution through green computing (Table 3).

**Table 3.** Sustainable agriculture.

| Components | Definition/Meaning | Characteristics |
|---|---|---|
| ICTs Infrastructure and resources sustainability | The ability to maintain digital systems (hardware and software) and human resources (such as IT specialists, services providers and data collectors) for long-term services to farmers. | Regular maintenance<br>Hardware replacement<br>Software upgrades<br>Budget for human resources and service providers<br>Energy consumption<br>Environmental impact of production and disposal of ICT hardware |
| Economic sustainability | Refers to a long-term increased farm production that eventually increases farmers' income. | Less input cost<br>High production<br>Good market price<br>Increased farmers' income |
| Environmental sustainability | Refers to actions taken consistently for conservation ecology by minimizing harmful agriculture and ICTs' environmental impacts. | Less use of agrochemicals<br>Use of fortified agrochemicals<br>Use of renewable energy<br>Energy-efficient hardware<br>Use of recyclable hardware<br>Less carbon emission from data centers |

Source: Author's compilation.

The literature review provides current status digital technology for sustainable agriculture, services available for smallholder farmers and Tanzania's case. Most established digital services for smallholder farmers lack environmental sustainability and sustainability of the infrastructure and resources that support the services. Some of the services in developing countries propose charging farmers and other beneficiaries to achieve sustainability of the services. For instance, the farmer profiling platform business model suggests that service providers receive revenue through interest paid on credit by farmers, commission on farm inputs and fees charged from buyers of the farm produce [16]. Although the model may achieve sustainability of the digital services, the burden cost is primarily on farmers, limiting the economic sustainability of individual farmers and farmers' organizations. Table 4 presents the availability of general digital transformations and agriculture sustainability to smallholder farmers.

*3.3. Digital Technology and Tanzanian Agriculture*

The Tanzanian government has consistently supported smallholder farmers and the agriculture sector. Since the 1960s, the government introduced 16 National Agriculture Input Voucher Systems (NAIVS) for farmers to access and use modern farm inputs (seeds and fertilizers) through contracted agro-dealers for improved production and income [66]. However, due to lack of government control, cheating and fraud, contracted agro-dealers sell the subsidized inputs at full market price, leading to deficient programs' impact on farmers [67].

Indeed, since the adoption of ICTs in the national development plans in 2003, many ICTs related projects have been conducted to address various challenges in the agricultural sector. Generally, the target areas are agricultural information dissemination by agricultural research institutions (ARIs) and extension services to farmers and farmers organizations (FOs) [13,14]. The increased use of mobile technologies also triggered projects on mobile farm services such as Global System for Mobile Association (GSMA) "Mobile for Development" projects and mobile applications to support farmers in different value chains [14], mobile application for poultry farmers [68], and mobile decision support systems [69,70]. Furthermore, the design of farmers digital advisory service called "Ushauri"

to provide access to context-specific information from extension agents increases capabilities in decision-making and adaptation to changing environments [71]. These digital services don't meet the needs of a farmer's entire ecosystem; nor are they sustainable, as some of the mentioned services don't exist due to lack of sustainability plans or because farmers do not use the service. Digital technology intervention could attenuate the challenges and improve smallholder farmers' access to services for increased production and income. Table 5 presents the summary of existing digital artifact solutions and services addressing some challenges of farmers in Tanzania.

**Table 4.** Digital services, smallholder farmers and agriculture sustainability.

| Literature | | Availability to Smallholder Farmers | Digital Technology and Agriculture Sustainability | | | |
| --- | --- | --- | --- | --- | --- | --- |
| | | | ICTs Infrastructure and Resources Sustainability | Economic Sustainability | Environmental Sustainability | |
| | | | | | Conservation Ecology | Green Computing |
| Digital technology for farm management | Data collection—IoT [6,27,29,31,32,66] | ✗ | ✗ | ✓ | ✓ | ✗ |
| | Data management and analysis [7,23,36] | ✗ | ✗ | ✓ | ✓ | ✗ |
| | DSS and VRT [30,31,33,34,37,38,40,41,69] | ✗ | ✗ | ✓ | ✓ | ✗ |
| Digital farmer profiling platform | [10,15,16] | ✓ | ✓ * | ✓ * | ✗ | ✗ |
| Agriculture sustainability | Economic sustainability [6–8,41,43,64] | ✗ | ✗ | ✓ | ✓ | ✗ |
| | Environmental sustainability — Conservation ecology [6–8,43] | ✗ | ✗ | ✓ | ✓ | ✗ |
| | Environmental sustainability — Green computing [65] | ✗ | ✗ | ✓ | ✓ | ✓ |

Source: Author's Compilation. Note: ✓ (Addressed) ✓ * (Addressed with limitations) ✗ (Not Addressed).

**Table 5.** A summary of digital services to farmers in Tanzania.

| Services | Problems | Digital Artifact Solutions | Sources |
| --- | --- | --- | --- |
| Financial | Lack of access to credit | None | [19,72] |
| Farm inputs | Counterfeit fertilizers, pesticides and herbicides | Agro-inputs Products Verification System (APVS) mobile application | [73] |
| Market | Access to market and market information | mFarming mobile service | [58,74] |
| Agriculture knowledge and information for decision making | Lack of information, farming knowledge and extension services | mAgri tracker GSMA Mobile for Development projects | [14] |
| | | Android mobile application for poultry farmers | [68] |
| | | A web and Mobile-Based Farmers' Advisory System for extension services | [53,60] |
| | | A mobile Decision Support System for access to climatic information | [69] |
| | | A mobile and web-based extension support system for horticulture farmers | [70] |
| | | "Ushauri" digital advisory service | [71] |

Source: Author's compilation.

Despite all the efforts, smallholder farmers in Tanzania still face many challenges in accessing services from other actors in a farmer ecosystem. Challenges include access to credit [19,72–76], substandard agricultural inputs from uncertified agro-dealers [77–80], unfair market prices due to the involvement of middlemen and lack of government oversight [14,57,81,82].

## 4. Discussion

This paper emphasized the digital technology and services in agriculture, focusing on the smallholder farmers' participation in sustainable agriculture. So far, similar to other sectors such as manufacturing industries, agriculture sector is undergoing major digital transformations through the application of cutting-edge digital technologies.

Inequalities: It is also important to note that digital transformations in agriculture are highly characterized by digital inequalities between large- and small-scale farmers, and between high-income and low-income countries. Governments, researchers, organizations and other stakeholders need to address factors leading to digital inequalities for smallholders to engage into sustainable agriculture. Sustainable agriculture by smallholder farmers require digital solutions for solving common challenges, which need strong commitment and collaboration among agricultural stakeholders at a country level, and then the adoption of advanced digital solutions such as precision technology.

Technology advancements create possibilities for solving many social-economic challenges that the world faces. Smart agriculture is the latest technology that uses the most advanced tools and software such as remote sensing, big data, IoT, information systems, AI, decision support system (DSS) and variable rate application (VRA) in farm management [6,7,41]. However, these digital advancements in agriculture are not equally available around the globe due to different social-economic factors. While developed countries are fast-moving in cutting-edge agricultural technologies (agriculture 4.0), developing countries are lagging, leading to low production and environmentally unfriendly practices [75,76]. Some of the developing countries are making steps towards precision agriculture. For instance, Bangladesh's online fertilizer recommendation system (OFRS) enables smallholder farmers to efficiently apply fertilizer for sustainable agriculture production [77]. A review study shows opportunities for adopting precision agriculture by smallholder farmers in Sub-Saharan Africa (SSA). Nevertheless, these technologies are mostly experimental and mainly used by large-scale commercial farms in few SSA countries [78].

ICT—an environmental concern: The uneven adoption of new technologies in agriculture affects more than one billion smallholder farmers worldwide, which the FAO considers the world's largest food producer by 70% [16]. Nonetheless, precision agriculture is challenged by the environmental sustainability issues caused by ICT. Therefore, green computing—"maximizing the efficiency of computing resources and minimizing environmental impact" [79] could be more useful to smallholder farmers due to its reduced costs and economic and environmental sustainability. The most established digital agriculture services have sustainability issues and exclude smallholder farmers.

Challenges: Despite the promising developments in technology, digital services in agriculture are yet to achieve complete sustainability. As the latest digital transformation, precision agriculture lacks the component of green computing, causing environmentally unfriendly practices. Precision agriculture is also poorly adopted by farmers, especially in developing countries, leaving most smallholder farmers behind in sustainable agriculture, in addition to the fact that, in general, small farms produce proportionally more greenhouse gas emissions than very large ones [80]. Furthermore, ICT infrastructure and resources sustainability are fundamental components for long-term agricultural production and profitability.

Profiling platform: A digital farmer profiling platform business model was recently designed to enable smallholder farmers' access to different services for increased production and income [10,15,16]. The model could achieve economic sustainability, but service

providers charging smallholder farmers directly and indirectly for infrastructure and resources sustainability affect farmers' profit margins. Lack of government participation in the model could lead to unsolved smallholder farmers challenges to some countries where government, for example, should control market price and subsidies to targeted poor farming communities. Digital farmer profiling also lacks environmental sustainability components.

Summary: Many large-scale farmers such as commercial farmers, wholesalers, traders and exporters have long invested in the use of ICT with well-developed farm inputs and market functions. For instance, precision agriculture uses advanced technology such as farm management information systems (FMIS), social networks and other complex customer and farm management systems [55]. Therefore, large-scale farmers are not often confronted with the sustainability of ICT infrastructure and resources as they cooperate in the business plan for investment. Digital services for smallholder farmers usually are established by the stakeholders such as the government, donors, commercial service providers, scientists and public–private partnerships; thus, the modality requires a proper mechanism for sustaining the infrastructure and other resources supporting the services.

Furthermore, the literature places more emphasis on economic and less onenvironmental sustainability. Engineers should also prioritize green computing when developing digital services for ecological sustainability in agriculture. The current digital technology systems in smart farming use cloud computing model to manage voluminous data through data centers. However, the data centers are highly wasteful in terms of expenses, energy consumptions and carbon emissions [81]. Furthermore, ICT hardware has an immense effect on the environment throughout its life cycle. The manufacturing phase involves using rare earth metals extracted under unfavorable environmental practices, which causes water, soil and air pollution, with high energy consumption in the use phase and e-waste produced in the final phase [82]. The cloud computing model commonly used in precision agriculture has also an immense negative impact on the environment due to carbon emission from data centers that host massive data [65]. To achieve the component of environmental sustainability, engineers propose using energy-efficient hardware, using renewable energy such as solar and wind, recycling e-waste and designing new tools such as cooling systems and datacenters with minimal impact to the environment [79,82–84]. We acknowledge the environmental impact of solar panels in their production and disposal phases; however, our focus is on the usage phase.

## 5. Conclusions

### 5.1. Literature Summary

This article provides an overview of the current status of digital technology and services available in agriculture sector, their contribution to sustainable agriculture and relationship to smallholder farmers. The digital technology varies from simple mobile and web-based applications, mostly for smallholders to complex autonomous, information and cyber-physical systems used by large scale farmers. Digital transformation seems promising and changing all aspects of life in different disciplines, leading to new business models, services and products. The use of digital technology in agriculture may solve the challenge of food insecurity in yet a constant population increase in the world. The literature analysis has shown that sustainable agriculture is a reality through digital technology and services. However, the cutting-edge digital technology in agriculture (smart farming) is not accessible to smallholder farmers who, despite their small size, produce over 70% of the world's food. The existing digital models for smallholder farmers, including the Tanzanian case, lack vital components of sustainable agriculture. They mainly address the needs of smallholder farmers at a particular stage of a farming cycle, such as farm preparations. Furthermore, the services are primarily for a specific country and crop value chain; examples are Tanzania (Table 3) and Kenya's KALRO mobile applications for different crops and livestock.

We found that the literature relates sustainable agriculture more with the precision technology. However, is it always needed, especially real-time precision agriculture, or sustainability can be achieved with other means? For instance, smallholder farmers are often reluctant in adopting precision technology even in developed countries [85–92], where management differs greatly between large and small farms. Perhaps establishing advisory services specifically for smallholder farms can be more efficient than using precision technology that communicates directly with the producer. Indeed, we believe that if smallholder farmers can access financial services (credit and insurance), quality farm inputs, subsidies, advisory services and market, they can increase production and profitability, adhere to environmentally friendly farming practices hence sustainable agriculture. Therefore, organizing agriculture stakeholders (including the government) at a country level and developing digital solutions that address common challenges of smallholder farmers could lead to sustainable agriculture and adoption of precision farming in developing countries. The limitation of this study is emphasized in identifying smallholder challenges towards sustainable agriculture in Tanzania case and proposing digital solutions. The needs of smallholder farmers may differ among countries and could need a thorough study to adopt the proposed digital solutions in a particular country's context. Additionally, the study focused more in crop farming, thus, did not cover digital technologies used for instance in livestock management.

*5.2. Towards a Comprehensive Digital Platform for Sustainable Agriculture in Smallholders Farms*

In the future, we plan to design and implement a digital platform for smallholder farmers to access all essential services (subsidies, credit, insurance, government services, market and farming information) under one roof. The platform will address the needs of smallholder farmers in a complete farming cycle—from farm preparations, farm inputs, harvesting and post-harvesting activities by consolidating agriculture stakeholders at a country level. The platform will also adhere to all critical components of sustainable agriculture, namely the sustainability of digital infrastructure and resources offering the services, economic and environmental sustainability.

**Author Contributions:** Conceptualization, G.E.M., G.D.M.S. and P.-Y.B.; methodology, G.E.M.; validation, G.D.M.S. and P.-Y.B.; formal analysis, G.E.M.; writing—original draft preparation, G.E.M.; writing—review and editing, P.-Y.B. and G.S; supervision, G.D.M.S. and P.-Y.B.; funding acquisition, G.E.M. All authors have read and agreed to the published version of the manuscript.

**Funding:** The APC was funded by University of Geneva.

**Institutional Review Board Statement:** Not applicable.

**Informed Consent Statement:** Not applicable.

**Data Availability Statement:** Not applicable.

**Acknowledgments:** This paper was part of the larger study supported by the ESKAS—Swiss Government Excellence Scholarship.

**Conflicts of Interest:** The authors declare no conflict of interest.

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
