# Peer review of "Digital Technology and Services for Sustainable Agriculture in Tanzania: A Literature Review"

_sustainability, doi:10.3390/su14042415_

Round 1
Reviewer 1 Report
The topic of the interdependencies between digital technologies (digitization) and sustainable farmings may regarded as important for futher empirical studies. This refers both to developing and developed countries. Exploring how digital technologies may affect sustainability of agricultural sector has value added for publc policies. As the author(s) have convicingly stated (ll. 55-57, p. 2), "This paper is part of a larger research project to harness digital technology for sustainable agriculture in Tanzania, which aims to identify knowledge gaps on digital
technology and services available to smallholder farmers and sustainability in agriculture".
The remarks:
Strengths
1. The transparent introduction with a set of research question.
2. A correct selection of articles/papers for lit. review.
3. The style of description is typical of research reviews.
The major shortcomings
As part 2.
1/ Could you defend your methodological attitude related to using a relatively simplified method of research review? In my opinion, the systematic review or review with elements of bibliometric analysis would be more beneficial.
2/ You should justify the order of presenting paragraphs in main body of paper.
3/ The number of papers (about 70) is not convinicing and seems to be rather limited. ICT tools have affected risk management in agriculture (incl. index-based contracts). This part seems to be very limited and even simplified.
4/ Point 5 related concluding remarks should be strengthened. The two short paragraphs are significant shortcomings of the manuscript.
To conclude, the paper should be reconsidered after a revision. The revised paper may attract a lot of potential readers, in particular researchers who deal with agricultural finance or risk management in agriculture.
Author Response
- We have now mentioned that we used the PRISMA guideline (updated version by McKenzie et. al., 2020) as a standard protocol and evidence-based framework for conducting review studies. We have added a new paragraph to provide the details of the boolean logics used to filter the literature and also inserted a Table showing identification, screening and inclusion of reports as per PRISMA guidelines.
- We have improved the manuscript as suggested.
- We have added a paragraph that shows the importance of insurance for smallholders particularly due to risks in production and the market (Index-based agriculture insurance has been an effective option in developing countries).
- The conclusion section was extended by summarizing the main gaps found in the literature, suggesting solutions for smallholder farmers, and pointing out the limitations of the study.
Reviewer 2 Report
Dear author,
The topic of the article is quite interesting and topical. It is an emerging topic in the field of Digital Economy and Agricultural Economics. Nevertheless, your paper has important weaknesses that should be corrected. I would like to point out some aspects to improve your manuscript. I explain my concerns in more detail below. I ask the authors to specifically address each of my comments in their responses.
Major comments
1) STRUCTURE OF THE PAPER. It is recommended that the structure of the paper be adjusted to make the content flow better and to improve the coherence of the research work provided. In particular, the following structure is recommended: 1) Introduction 2) Method 3) Results 4) Discussion and 5) Conclusions. The discussion of results is particularly relevant. Authors should discuss the results and how they can be interpreted from the perspective of previous studies and of the working hypotheses. The findings and their implications should be discussed in the broadest context possible.
2) INTRODUCTION: authors are encouraged to include a paragraph describing the original contribution of their study and its contribution to the advancement of knowledge.
In this section, it is recommended to complete the background research review with other previous studies that have analysed digitisation in different sectors and territories. In other words, it would be advisable to include a few paragraphs in the introduction to contextualise the phenomenon of digitisation. We suggest some current references that may help you to complete this review:
-Muhamad, S., Kusairi, S., Man, M., Majid, N. F. H., & Kassim, W. Z. W. (2021). Digital adoption by enterprises in Malaysian industrial sectors during COVID-19 pandemic: A data article. Data in Brief, 107197. https://doi.org/10.1016/j.dib.2021.107197
-Jorge-Vázquez, J.; Chivite-Cebolla, M.P.; Salinas-Ramos, F. The Digitalization of the European Agri-Food Cooperative Sector. Determining Factors to Embrace Information and Communication Technologies. Agriculture 2021, 11, https://doi.org/10.3390/agriculture11060514
-Massaro, M. (2021). Digital transformation in the healthcare sector through blockchain technology. Insights from academic research and business developments. Technovation, 102386. https://doi.org/10.1016/j.technovation.2021.102386
-Mohapatra, B., Tripathy, S., Singhal, D., & Saha, R. (2021). Significance of digital technology in manufacturing sectors: Examination of key factors during Covid-19. Research in Transportation Economics, 101134. https://doi.org/10.1016/j.retrec.2021.101134
3) METHODOLOGY: the authors should make a notable effort to describe more precisely the methodology used in their study. The authors propose a bibliometric analysis, however, they don't specify the method used. Nor is the method validated from the perspective of previous studies.
4) CONCLUSIONS: these should be significantly improved. Conclusions should be related to the objective of the research. This section should indicate the result of the bibliometric analysis carried out and, therefore, the state of research in this field. It would also be advisable to indicate the limitations of the study.
Minor comments:
1) The format of the text should be uniform. There are paragraphs with different line spacing and font size.
2) The references in the text should be revised and adapted to the format of the journal. For example, [13-16] instead of [13]-[16].
3) REFERENCES: references must conform to the style standards proposed by the journal:
References must be numbered in order of appearance in the text (including citations in tables and legends) and listed individually at the end of the manuscript. We recommend preparing the references with a bibliography software package, such as EndNote, ReferenceManager or Zotero to avoid typing mistakes and duplicated references. Include the digital object identifier (DOI) for all references where available.
Citations and references in the Supplementary Materials are permitted provided that they also appear in the reference list here.
In the text, reference numbers should be placed in square brackets [ ] and placed before the punctuation; for example [1], [1–3] or [1,3]. For embedded citations in the text with pagination, use both parentheses and brackets to indicate the reference number and page numbers; for example [5] (p. 10), or [6] (pp. 101–105).
-
- Author 1, A.B.; Author 2, C.D. Title of the article. Abbreviated Journal Name Year, Volume, page range.
- Author 1, A.; Author 2, B. Title of the chapter. In Book Title, 2nd ed.; Editor 1, A., Editor 2, B., Eds.; Publisher: Publisher Location, Country, 2007; Volume 3, pp. 154–196.
- Author 1, A.; Author 2, B. Book Title, 3rd ed.; Publisher: Publisher Location, Country, 2008; pp. 154–196.
- Author 1, A.B.; Author 2, C. Title of Unpublished Work. Abbreviated Journal Name stage of publication (under review; accepted; in press).
- Author 1, A.B. (University, City, State, Country); Author 2, C. (Institute, City, State, Country). Personal communication, 2012.
- Author 1, A.B.; Author 2, C.D.; Author 3, E.F. Title of Presentation. In Title of the Collected Work (if available), Proceedings of the Name of the Conference, Location of Conference, Country, Date of Conference; Editor 1, Editor 2, Eds. (if available); Publisher: City, Country, Year (if available); Abstract Number (optional), Pagination (optional).
- Author 1, A.B. Title of Thesis. Level of Thesis, Degree-Granting University, Location of University, Date of Completion.
- Title of Site. Available online: URL (accessed on Day Month Year).
All in all, this is very promising research. I hope these comments will be helpful.
Best regards,
Author Response
We thank you for your wonderful comments and suggestions.
- We have adapted the recommended structure in our paper and improved the discussion section as suggested. We adjusted some of the content (e.g Table 3) by moving them where they were previously located to fit into this new structure.
-
The concept of digitization has been discussed showing its impact in different sectors and factors for its rapid growth such as increased interconnectedness and the pandemic breakout.
- We have now mentioned that we used the PRISMA guideline (updated version by McKenzie et. al., 2020) as a standard protocol and evidence-based framework for conducting review studies. We have added a new paragraph to provide the details of the boolean logics used to filter the literature and also inserted a Table showing identification, screening and inclusion of reports as per PRISMA guidelines.
-
We have significantly improved this part to the best of our knowledge, considering all the comments suggested.
- We also worked on the minor comments references in particular.
We downloaded and installed the Sustainability citation style to our Mendeley Desktop software to automatically take care of the in-text citation and reference list.
Reviewer 3 Report
The article is a review paper aiming at identifying knowledge gaps on digital technologies and services for sustainable agriculture and their availability to small farmers. The authors set at first a worldwide goal and then focusing on the Tanzania’s case. As compared to other review papers it is interesting because it includes grey literature that is not found in standard scientific databases.
The abstract could be more informative. For instance, it could include the main conclusions of the review accomplished. The problem is that the conclusions section is too reduced for the review work that has been done.
The Research Methods section needs some improvement. The review starts with an large-scale database search (1981 articles, line 92) and finishes analyzing 36 articles, 12 of them about the Tanzania’s case. Maybe including a table explaining the main features of the articles finally reviewed could improve the reader’s overall understanding of the work presented. Please refer to PRISMA (http://prisma-statement.org/) to improve this section.
The first paragraph of section 3.3 (lies 371 to 380) belongs to the “Methods” section
Conclusions section is too reduced. The authors need to summarize the gaps found in the literature review. For instance, related to crops or farming systems relevant to Tanzania’s case. If this work is the background for a future project implementation it should include a detailed description of the agricultural system in Tanzania, main agro-climatic areas, main crops, main livestock, cropped areas and percentages of final agricultural output, for instance.
Minor spelling or format corrections:
Line 92: Substitute “Elservier” by “Elsevier”.
Line 93: Substitute “Mendely” by “Mendeley”.
Line 148 plasmopara viticola should be Plasmopara viticola (in italics).
Author Response
Dear Sir/Madam
We thank you for your wonderful comments and suggestions.
- The abstract has been improved, highlighting significant results of the review and the main points of the conclusion as suggested.
-
We have inserted a table showing the PRISMA flow of reports "identification", "screening" and "included" for both general literature and Tanzania’s case, making a total of 36 reports.
-
We have moved the paragraph to the methods section as suggested.
- The conclusion is now improved, summarising major gaps found in the literature and showing the study's limitations. However, we couldn’t find a better way to include Tanzanian case details such as crop, livestock, etc. Our study focused on reviewing how digital technologies are used and how they address smallholder farmers' needs towards sustainable agriculture. The suggested inputs are beneficial in our future project, which will deal more specifically with the Tanzanian case and how the proposed solution fits the country's agricultural system.
-
We have done all the minor corrections suggested.
Reviewer 4 Report
Overall, an interesting contribution to the challenges of adopting precision agriculture and the digital transformation in developing countries and smallholder farmers. This issue is often neglected as precision agriculture is generally reviewed for developed or emerging countries such as Brazil and India. However, I was not able to find the suggestions for improvements by the authors mentioned in the abstract (these suggestions should be also briefly described in the abstract). This is unfortunate as I believe the entire relevance of this literature study lays on these recommendations elaborated by the authors.
Some points are not clear or do not seem relevant at all. For instance, the authors cover at great length the issue of "green computing" (an entire paragraph), while this is not specifically an issue for smallholder farmers in Tanzania but rather an issue for any country or agricultural systems.
More in line with sustainability would be the issue of greenhouse gas emissions as low production systems often emit larger amounts per unit of output. Precision agriculture can help here with the decision making but this might be different for smallholder farmers in Tanzania. In addition, animal welfare as an aspect of sustainability has not been covered despite the huge advancements made when relying on precision farming. Its potential for smallholder farmers in developing countries are yet to be evaluated.
I am also missing somewhat a more general reflection on whether precision technology, especially real time precision agriculture, is always needed or if sustainability can be achieved with other means? For instance, smallholder farmers are often reluctant in adopting precision technology even in developed countries where management differs greatly between large and small farms. Perhaps establishing advisory services specifically for smallholder farms can be more efficient than using precision technology that communicates directly with the producer?
Author Response
Dear Sir/Madam
We are thankful for your wonderful comments and suggestions.
- The suggestions for improvements have now been mentioned in the abstract, discussion and conclusion parts.
-
To answer your comment, we also added a reference that proposes a new environmentally “friendly” model for precision agriculture that differs from the more common cloud computing model. We also make it clear that it is not an issue specific to Tanzania. Qureshi, S. H. Mehboob, and M. Aamir, “Sustainable Green Fog Computing for Smart Agriculture,” Wirel. Pers. Commun., vol. 121, no. 2, pp. 1379–1390, 2021, doi: 10.1007/s11277-021-09059-x.
-
Greenhouse gas emissions as such are not the topic of this review, nor animal welfare. We focus more on digital technology in the agricultural sector. However, we added in the discussion part about green computing a mention that small farms proportionally might emit more gas than very large farms, with a reference supporting this fact.
- In our article we found that the literature relates sustainable agriculture more with precision technology, however, looking at the suggested solutions for improvements for smallholder farmers you will find that sustainable agriculture could be achieved by other means. When smallholder farmers have access to financial services (credit and insurance), access to quality farm inputs, subsidies, advisory services and markets can increase production and profitability, adhere to environmentally friendly farming practices hence sustainable agriculture. This could also trigger the adoption of precision technology by smallholder farmers in developing countries. We now address this point in the conclusion part, which takes into account such a reluctance to precision technology.
Round 2
Reviewer 1 Report
The manuscript has been strongly improved. My main suggestions have been implemented. The paper may be attractive for several group of readers. I recommend accept in a present form.
Reviewer 2 Report
Dear authors,
I think the authors have made a remarkable effort to improve the article. They have improved the structure and organisation of the content. On the other hand, the review of the research background has been suitably expanded. The authors have also made an effort to improve the description of the methodology used and the limitations of their study. The conclusions were also improved.
Wish you all the best. Best regards,